# *Aeromonas* Species Diversity, Virulence Characteristics, and Antimicrobial Susceptibility Patterns in Village Freshwater Aquaculture Ponds in North India

**DOI:** 10.3390/antibiotics14030294

**Published:** 2025-03-12

**Authors:** Alka Nokhwal, Rajesh Kumar Vaid, Taruna Anand, Ravikant Verma, Rachna Gulati

**Affiliations:** 1National Centre for Veterinary Type Cultures, ICAR-National Research Centre on Equines, Sirsa Road, Hisar 125 001, (Haryana), Indiataruna.anand@icar.gov.in (T.A.); 2Department of Zoology and Aquaculture, College of Basic Sciences and Humanities, Chaudhury Charan Singh Haryana Agricultural University, Hisar 125 004, (Haryana), India; 3College of Fisheries Science, Chaudhury Charan Singh Haryana Agricultural University, Hisar 125 001, (Haryana), India

**Keywords:** *Aeromonas*, antimicrobial resistance, *gyr*B gene, β-lactamases, aquaculture, India

## Abstract

Background/Objectives: Motile aeromonads are ubiquitous aquatic Gram-negative opportunistic pathogens with environmental, animal, aquatic, and human health implications. Methods: Motile aeromonads were isolated from village pond water samples (*n* = 100) of the Hisar district of Haryana state in India. Selective isolation and enumeration were followed by biochemical and genotypic identification using *gyr*B gene; evaluation of seven putative virulence factors and antimicrobial resistance studies and determination of extended spectrum beta lactamase (ESBL) and AmpC beta lactamase (ACBL) enzyme-producing abilities took place. Results: The viable counts of motile aeromonads varied from 1.6 × 10^2^ CFU/mL to 1.2 × 10^8^ CFU/mL. Six species of *Aeromonas* were identified with high prevalence of *A. veronii* (74.7%), followed by *A. caviae* (8.9%), *A. hydrophila* (7.6), *A. jandaei* (5%), *A. sobria* (2.5%), and *A. dhakensis* (1.3%). PCR amplification of seven genes related to virulence indicated that the majority of the isolates were positive for enolase (*eno*, 98%), cytotoxic enterotoxin (*act*, 88%), and hemolysin (*asa*1, 86%). Many isolates were also positive for type III secretion system inner membrane component (*asc*V, 53%), ADP-ribosylating toxin (*aex*T, 47%), and extracellular hemolysin (*ahh*1, 4%). The antimicrobial resistance (AMR) profile of the isolated *Aeromonas* isolates indicated the high resistance observed to nalidixic acid (40.2%), cefoxitin (33%), and imipenem (6.2%). In addition, the occurrence of 10.3% ESBL, 32% ACBL, and 29.9% multi-drug resistant (MDR) isolates is alarming. Phylogenetic analysis of *gyr*B sequences of *A. veronii* isolates (*n* = 59) together with GenBank sequences of *A. veronii* from different geographical regions of the world indicated high genotypic diversity. Conclusions: the village aquaculture ponds in Hisar district have a high occurrence of MDR *A. veronii*, *A. hydrophila*, and *A. caviae*, posing significant animal and public health concern.

## 1. Introduction

Members of the genus *Aeromonas* are Gram-negative, rod-shaped, facultatively anaerobic bacteria that are unique to aquatic environments [1]. Many common species are considered to be emerging pathogens that not only infect aquatic organisms such as fish, but also act as opportunistic pathogens for humans and animals [2]. Given their abundance in freshwater lakes and ponds, they are of immense economic importance in pisciculture. Diseases such as furunculosis, hemorrhagic septicemia or motile *Aeromonas* septicemia, hemorrhagic enteritis, epizootic ulcerative syndrome, red sore disease, tail and fin rot, and lethargy are reported in fish [3]. A few of the important *Aeromonas* spp., viz., *Aeromonas caviae*, *Aeromonas veronii*, *Aeromonas hydrophila*, and *Aeromonas dhakensis*, are frequently associated with opportunistic infections, such as septicemia, gastroenteritis, and cutaneous infections, in immunocompromised individuals [4]. Therefore, the identification of *Aeromonas* spp. is important for understanding disease ecology.

However, the identification of motile aeromonads has its own set of intricacies. The motile aeromonads group is composed of 32 species and 12 validly published subspecies [5]. Formerly placed in the family *Vibrionaceae*, the genus *Aeromonas* taxonomy was revised in 1986 to create a new family, *Aeromonadaceae*, based on the results of 5S rRNA, 16S rRNA, and rRNA-DNA hybridization approaches [6]. However, phenotypic biochemical methods as well as molecular 16S rRNA processing pose challenges and are not sufficiently discriminatory for species identification due to the variable behavior of motile aeromonad strains [7,8,9]. Due to existing heterogeneity within the group, the phylogeny of housekeeping genes, mainly that of *gyr*B, a type II DNA topoisomerase encoding the β-subunit of DNA gyrase, has been considered an appropriate indicator for bacterial systematics [9,10].

The pathogenicity of *Aeromonas* spp. is a function of its virulence-encoding genes, which assist the bacterium in colonizing, overwhelming the host immune system and establishing infection. These include various extracellular enzymes, lipopolysaccharides (LPS), pore-forming aerolysin, hemolytic toxins, cytotoxic and cytotonic enterotoxins, the type III secretion system (T3SS), and associated effectors [3,11]. Therefore, in view of the increased one health significance, it is important to understand the genotypic basis of these factors for characterizing the pathogenic potential of aeromonads.

Freshwater fish farming, arguably considered to be a sustaining component of the aquaculture industry, is thought to support income and employment in developing countries [12]. Bacterial infections associated with poor water quality are some of the challenges confronting freshwater aquaculture. For a long time, antimicrobial agents and chemotherapies have been primarily employed to control bacterial diseases caused by aeromonads [13]. The most widely utilized antimicrobials in the aquaculture industry are tetracyclines, enrofloxacin, erythromycin, amoxicillin, sulfonamides, oxolinic acid, oxytetracycline, chloramphenicol, and florfenicol [14]. The lack of uniform antimicrobial standards in aquatic environments is the prime reason for their frequent overuse and exploitation in fish culture [15], especially because antimicrobials may be easily available as over-the-counter medications and may be used off label [16]. Antimicrobial residues may be retained in fish fecal matter and uneaten food, where subsequent entry of these antibiotics into the aquatic environment and selective genome mutations in bacteria can lead to antimicrobial resistance (AMR). In this regard, motile aeromonads, ubiquitous in natural water bodies, can be considered good indicators of antimicrobial resistance, water quality, and sewage pollution, which has been well documented by various studies [17,18,19,20].

The water quality of natural ponds deteriorates due to organic and inorganic sewage effluent, defecation by domestic animals, solid waste disposal by locals, and poor drainage and mismanagement. Deterioration in water quality can be indicated by specific physicochemical parameters, which predispose the fish towards infection by such opportunistic pathogens [21]. Among these parameters, the most important are increased pH, total dissolved solids (TDS), temperature, salinity, turbidity, water electrical conductivity (EC), and decreased dissolved oxygen (DO) [16,22].

Our study highlights the importance of inland aquaculture pond systems as the ecological compartments of the environment, which, on account of their role as endpoint of organic wastes from humans and animals, may act as local epicentres of AMR evolution and dissemination, affecting the integrity of the aquaculture food chain, as well as animal and human health. The world is in the midst of a silent pandemic of antimicrobial resistance. The global action plan on AMR emphasized the gathering of evidence of an antimicrobial resistance base through surveillance and research. This study aims to address the aquatic environmental surveillance on AMR in village pond waters used for aquaculture, animal husbandry, and humans in the northern region of Haryana (India) for the first time. Therefore, to understand the prevalence of motile aeromonads in village pond waters used for aquaculture in this region, their species-level identification and antimicrobial resistance profile were investigated in natural freshwater village ponds in/around the Hisar district (Haryana, India). The physicochemical parameters of the pond water samples were evaluated. The *Aeromonas* spp. bacterial isolates were identified by phenotypic and biochemical testing and at the molecular level using *gyr*B sequencing. The genetic traits associated with the virulence of motile aeromonads isolated from pond water were also evaluated. The identified isolates were subjected to antimicrobial resistance profiling using a set of antimicrobial agents.

## 2. Results

One hundred pond water samples were collected from 66 village aquaculture ponds in the regions in/around Hisar (Haryana, India). Physicochemical characterization was performed on the water samples, and aeromonad bacteria were isolated from the water samples.

### 2.1. Water Quality Parameters

The physicochemical water quality parameters were measured in the examined pond water samples (*n* = 100), for which the range obtained is given in Table 1. The parameters of temperature (29–34 °C), pH (6.8–9.0), salinity (0.1–0.3 ppt), and dissolved oxygen (2.9–8.0 ppm) did not exhibit marked variations from the reference range [23,24,25]. Two of the evaluated parameters, TDS and EC, exceeded the desired values for some ponds, with the highest values being 2440 ppm and 3641 µs/cm, respectively.

### 2.2. Prevalence and Diversity of Aeromonas Species in the Fish Culture Ponds

Large, smooth, and honey-yellow colonies on SAA consisting of Gram-negative, pleomorphic rod-shaped bacteria were tentatively identified as *Aeromonas* species [26]. A total of 100 Gram-negative, oxidase-, and catalase-positive colonies were selected and isolated as putative members of the genus *Aeromonas* from all 100 pond water samples and cryopreserved as glycerol stocks until further analysis. The counts of motile aeromonads recovered using SAA varied from 1.6 × 10^2^ CFU/mL to 1.2 × 10^8^ CFU/mL in the sampled pond waters.

For phenotypic confirmation, these 100 isolates were subjected to further biochemical tests, and the presumed motile aeromonads exhibited reactions characteristic of the genus *Aeromonas* spp. (Appendix A). The isolates produced positive reactions to catalase, oxidase, and D-glucose, while variable reactions were observed using citrate, malonate, lactose, and sucrose. None of the presumptive aeromonads were able to produce H_2_S gas or urease enzyme. β- and α-haemolysis were demonstrated by 58.8% and 37.1% of the isolates, respectively, on 5% sheep blood agar (SBA), while only 4.1% of the isolates were non-haemolytic. Most of the isolates had fermentative properties (75.3%) and were all motile (88.7). The majority (99%) of the isolates could utilize glucose in the medium as their energy source, while only 36.1% of the isolates could metabolize lactose/sucrose present in the medium. Further, 69.1% of the total isolates showed gas production from D-glucose utilization. On the other hand, 56.7% and 7.2% of the isolates could use citrate and malonate as their carbon and energy sources, respectively. Finally, 87.6% of the isolates could breakdown the amino acid tryptophan into indole and were considered indole-positive (Appendix A).

For species-level identification, these isolates (*n* = 100) were further analyzed by PCR amplification of the housekeeping gene *gyr*B. BLAST analysis of the *gyr*B sequences revealed that 79 isolates were most closely related to *Aeromonas* genus members in the NCBI database (percent identity between species ranging from 98.04 to 100%). The *gyr*B sequences of 79 isolates were deposited in the NCBI GenBank database, and the accession numbers were assigned (Appendix A). The submitted sequences belonged to six different *Aeromonas* species (Figure 1), which included *A. veronii* (*n* = 59), *A. caviae* (*n* = 7), *A. hydrophila* (*n* = 6), *A. jandaei* (*n* = 4), *A. sobria* (*n* = 2), and *A. dhakensis* (*n* = 1). Additionally, some isolates were not grouped into any cluster with known *Aeromonas* species strains, and these isolates need further investigation using more than one gene or other approaches. Therefore, these isolates were provisionally considered as “*Aeromonas* spp.” (*n* = 18). In addition to *Aeromonas* spp., our *gyr*B gene sequencing also identified three isolates as *Stenotrophomonas maltophilia*, *Pseudomonas alcaliphila*, and *Pseudomonas sediminis* (one isolate each), which may belong to the normal fish microbiota or the aquatic ecosystem and are reported as fish pathogens under stress conditions [27]. Therefore, these three isolates were discontinued for the current study and only 97 isolates were taken up for further analysis. As an outcome of this survey, the phylogenetic relationships of the predominant species, *A. veronii* strains, were determined using *gyr*B gene sequences from 59 total Indian (NCVTC) isolates in this study, along with 19 other *A. veronii* strains from GenBank (representing various countries and hosts).

A consensus phylogenetic tree with bootstrap support was constructed to illustrate the evolutionary relationships of these *A. veronii* sequences based on a dataset of 1190 nucleotide positions (Figure 2). The evolutionary relationships of other *Aeromonas* species isolated from this study were also drawn (Appendix A). Nucleotide sequence analysis revealed that the *A. veronii* isolates were grouped into two highly conserved branches, corresponding to two major clades. One major clade, comprising 53 isolates, was divided into two distinct subgroups, suggesting a greater degree of genetic similarity among these isolates. The smaller subgroup (node ID 60) consisted of 19 isolates, primarily from India, along with one isolate each from Switzerland (I110/2016) and Egypt (Adaw220314/2022). This clustering pattern suggests that ecological niches, rather than geographical origin, are key determinants of evolutionary relationships among *A. veronii* isolates. In contrast, the larger subgroup (node ID 34) consisted of 34 isolates, including 20 from India and 14 from 10 other countries. One minor branch (node ID 46) grouped an Indian isolate with a U.S. isolate (H-1159/2013), possibly indicating a common origin. Another notable subgroup (node ID 6) clustered an Indian isolate (Aq55) with isolates from China, Australia, Spain, Korea, and Portugal, showing global clustering patterns. Isolates Aq88 and 5TK6 from India and Malaysia, respectively, branched with an Australian isolate (A21-4/2021) under node ID 58, while isolates Aq15 and Aq38 clustered with a Japanese isolate (AAr1218/2018) under node ID 43. A minor clade (node ID 94) consisted primarily of 25 Indian isolates, with only two isolates from other countries. For instance, isolates Aq31 and Aq24 clustered closely with a Japanese isolate (WL4-118/2015), while Aq12 formed a distinct branch with a Chinese isolate (CS-46/2012) under node ID 74. This analysis suggests close evolutionary distances among these isolates, highlighting the diversity within *A. veronii* species.

### 2.3. Virulence Gene Characteristics

Among the 97 genotypically confirmed *Aeromonas* isolates, a greater number of the isolates (34.02%; 33/97) were found to carry 4 out of 7 virulence genes, followed by 24.74% (24/97), 29.9% (29/97), 7.22% (7/97), and 3.09% (3/97) of the isolates carrying 5, 3, 2, and 1 screened virulence gene (s), respectively (Figure 3). One isolate of *A. hydrophila* (1.03%; 1/97) was detected to carry as many as 6 out of the 7 tested virulence genes.

On average, enolase (*eno*, 97.9%) was the most frequently detected virulence gene in *Aeromonas* species, followed by cytotoxic enterotoxin (act, 88.7%) and hemolysin (asa1, 85.6%), which had nearly similar detection frequencies (Table 2). *Asc*V (a type III secretion system inner membrane component) was detected at a lower frequency of 51.6%, followed by the occurrence of the ADP ribosylating toxin gene *aex*T at 45.4%. The least frequent gene was the extracellular hemolysin gene *ahh*1, with a 4.12% prevalence. In contrast, the heat-stable cytotonic enterotoxin gene *ast* was not detected in any of the 97 isolates.

The dendrogram obtained by clustering *Aeromonas* isolates on the basis of the presence or absence of different virulence genes was depicted by the formation of two different clusters (shown with different colors) (Appendix A) which further delineated into two major groups and two minor groups, revealing virulence-mediated heterogeneity in the species. Both the major groups were dominated by *A. veronii* isolates. Either the isolates were positive for *asa*/*act*/*asc*V/*eno*/*aex*T (shown in purple and red) or for *asa/act/eno* (shown in blue). One of the two minor clusters (shown in black) clustered together the *A. veronii* isolates positive for *asa*/*act*/*eno*/*aex*T genes. Furthermore, the fourth minor cluster (shown in green) was highly variable in composition, accounting for most of the *A. hydrophila* strains, along with the *A. caviae* and *A. jandaei* isolates. Most of the *A. hydrophila* and *A. sobria* isolates were found to harbor the virulence genes *asa*1/*ast*/*asc*V/*eno*, whereas all *A. jandaei* were positive for *asa*1/*asc*V/*eno*/*aex*T. Further, most of the *A. caviae* isolates were positive for *asa*1/*act*/*eno* and *A. dhakensis* isolate was found to harbor *asa*1/*eno* genes. The cluster dendrogram revealed the characteristic species-wise distribution of each *Aeromonas* species and that the different species belonged to different groups/clusters (e.g., *A. hydrophila*), with few exceptions where the number of isolates was low (e.g., *A. jandaei* and *A. caviae*) (Appendix A).

### 2.4. Antimicrobial Susceptibility Patterns

Antimicrobial susceptibility profiles of 97 isolates identified as *Aeromonas* species by the sequencing of the *gyr*B gene using 15 common antimicrobial agents (8 classes) was performed by disc diffusion assay (Figure 4). Variable susceptibility patterns to all the tested antimicrobial agents were recorded (Figure 4; Appendix A) except ampicillin (AMP), for which all the isolates were resistant. Further, all of the isolates were susceptible to norfloxacin (NOR). Resistance to imipenem was observed in five *A. veronii* isolates (8.47%) and one *A. caviae* isolate (14.29%). Five isolates (5.2%) were resistant to tetracycline (three *A. hydrophila*, one *A. veronii*, and one *Aeromonas* spp. isolate). Resistance to nalidixic acid (NAL) and cefoxitin (FOX) was observed in 40.2% and 33% of the total 97 isolates, respectively. Resistance to cefoxitin, which is a cephamycin, second-generation cephalosporin, was found in 83.3% *A. hydrophila* and 57.1% *A. caviae* isolates followed by 27.1% *A. veronii* isolates. Similarly, high resistance to nalidixic acid and synthetic quinolones was observed in 66.7% *A. hydrophila* isolates, followed by 57.1% *A. caviae* and 35.6% *A. veronii* isolates. Therefore, proportionally more *A. hydrophila* and *A. caviae* isolates than *A. veronii* isolates were resistant to cefoxitin and nalidixic acid, possibly because of the significant difference in the number of isolates. Further, all isolates expressed variable degrees of resistance to imipenem (IPM), cefotaxime (CTX), aztreonam (ATM), chloramphenicol (CHL), ceftazidime (CAZ), trimethoprim-sulfamethoxazole (SXT), tetracycline (TET), and amoxiclav (AMX) at frequencies less than 10%.

Furthermore, synergy-based examination of phenotypically and genotypically confirmed *Aeromonas* isolates for the production of extended-spectrum β-lactamase (ESBL) and AmpC β-lactamase (ACBL) was performed. The results reveal that in total, 10.3% isolates (*n* = 10) were extended-spectrum β-lactamase (ESBL) producers, comprising largely *A. veronii* (7.21%, *n* = 7), with *A. hydrophila* (5.15%), *A. caviae* (2.06%), *A. dhakensis* (1.03%), and others also included in the group. On the other hand, 34.02% isolates (*n* = 33) were ampC β-lactamase producers, dominated by *A. veronii* (16.5%, *n* = 16), followed by *A. hydrophila* (5.15%, *n* = 5) and *A. caviae* (4.12%, *n* = 4). Additionally, one *A. dhakensis* isolate was ESBL-positive, while no *A. jandaei* or *A. sobria* showed ACBL production.

The overall MAR index ranged from 0.07 to 0.73, with 29.9% of the isolates exhibiting a MAR index > 0.2 (Figure 5). Approximately 32.9%, 37.1%, 16.5%, 6.2%, and 3.1% of the isolates were resistant to 1, 2, 3, 4, and 5 out of the 15 tested antimicrobial agents, respectively, while 1.03% of the isolates were each resistant to 6, 7, 8, and 11 antimicrobial agents having a high MAR index (0.4–0.7). The majority of the isolates were resistant to either one (33%), i.e., ampicillin or two antimicrobial agents (37.1%), i.e., ampicillin and nalidixic acid. Nearly 16.5% of the isolates in this study were on the verge of becoming high-risk sourced isolates (MAR index = 0.2). On the other hand, the highest MAR index of 0.73 was observed for an *A. veronii* isolate (resistant to 11 antimicrobial agents).

## 3. Discussion

The genus *Aeromonas* consists of Gram-negative rods that are ubiquitous in all types of aqueous environments. Consequently, several species of this genus, viz., *A. hydrophila*, *A. salmonicida*, *A. veronii*, *A. jandaei*, *A. caviae*, and *A. bestiarum* have been described as important fish pathogens [28]. These factors therefore constitute an important negative input as disease-causing limiting factors in sustainable fish farming worldwide. In view of the aquatic and public health importance of motile aeromonads, this study surveyed the prevalence of *Aeromonas* spp. in natural freshwater aquaculture village ponds in the Hisar district (Haryana, India).

Motile *Aeromonas* spp. were isolated from all 66 village aquaculture pond water samples. The genus *Aeromonas* is among the dominant bacteria inhabiting pond ecosystems and is a good indicator for assessing water quality [29]. A study in Egypt reported *Aeromonas* isolates in 12.5% of water samples [21]. Similarly, in Trinidad and Tobago, *Aeromonas* spp. were isolated from 60% of pond water samples [30]. The genus *Aeromonas* can be used almost synonymously with aquatic environments [3]. However, an exceptionally high rate of isolation of *Aeromonas* spp. from sampled pond water in our study also indicates that either contaminated water is sourced into ponds or that ponds become contaminated locally. The incidence of *Aeromonas* in wastewater has been reported to be high; moreover, the concentration of *Aeromonas* may be associated with terrestrial water effluents [31]. A variety of biomolecules, including proteins and lipids originating from the decomposition of dead flora, fauna, detritus, connective tissues, animal faeces and urine, etc., are present in earthen freshwater ponds [17]. In the current study, all the ponds had a freshwater canal system as a water source apart from seasonal rainwater. However, ponds also received additional drainage water from nearby residential sources, but no agricultural or other runoff was reported. The surveyed ponds were also commonly used for livestock and anthropogenic activities. This may be the reason for the detection of *Aeromonas* spp. in all the ponds, since aeromonads are carried in the fecal matter of domestic animals [32]. All the surveyed ponds also served as wallowing sites for water buffaloes (*Bubalus bubalis*), which are important milch animals in this region, providing an easy gateway to constant exposure and interactions for the transmission of microbial contaminants and diseases, including those between aeromonads and humans. This finding is consistent with the strong associations of this genus with human infections [3]. They also reported that >85% of human *Aeromonas* infections are attributed to *A. hydrophila*, *A. caviae*, and *A. veronii* bv. *sobria* only. Researchers identified *Aeromonas veronii* as an established human pathogen [29]. This study, for the first time, reports the occurrence and identification of aeromonads in natural village aquaculture ponds in the sampled region of Haryana (India).

The quantitative data on the *Aeromonas* spp. inhabiting aquaculture ponds covered in this study varied from 1.6 × 10^2^ CFU/mL to 1.2 × 10^8^ CFU/mL. This level of *Aeromonas* corroborates (2.5 × 10^6^ CFU/mL) with the previous data reported by Skwor and coworkers [29] from water samples of Lake Erie, USA, but is higher (10^1^–10^3^ CFU/mL) than the values noted by other authors [33] from earthen culture ponds of tilapia in the Philippines and the number of *Aeromonas* (2.1–2.6 × 10^6^ CFU/mL) in the river water of Lotcha (West Bengal, India) [34]. A significant correlation has been reported between water temperature and total culturable *Aeromonas* spp. during the dry period at temperatures >25 °C [35]. Hisar district is located in an arid zone with a hot and dry climate. During most of the year (May to September), the average minimum diurnal temperature is >25 °C. In our study, which was performed during the months of June to Sept., the average water temperature was 32.3 °C (Table 1), which indicates a favorable environment for *Aeromonas* spp. growth and survival.

Most of the water physicochemical properties of all the ponds examined during the present study were within the range suitable for freshwater fish culture [23,24,25]. Furthermore, with these data, it can be inferred that a highest electrical conductivity observed for one sample (>2000 µS/cm) is still within the acceptable range (30–5000 µS/cm) [25]. However, monitoring the *Aeromonas* spp. load in such culture ponds is necessary, as are physicochemical parameters, which are integral components of pond management strategies for determining the quality of fish produced in ponds [33]. Sadique and workers reported that in addition to water temperature, no other physicochemical parameters (pH, DO, salinity, or TDS) were found to have any significant influence on the quantity of *Aeromonas* in water bodies [35].

We identified *Aeromonas* spp. isolates on the basis of *gyr*B sequence homology analysis in the NCBI database using the nBLAST tool. Consistent with previous findings, biochemical identification of aeromonads has limitations [1], as in the present study, a few phenotypically presumptively identified aeromonads were later identified as *Pseudomonas* spp. and *Stenotrophomonas* spp. through genetic *gyr*B characterization. The dominant species identified in this study included *A. veronii*, *A. hydrophila*, and *A. caviae*. This is similar to the findings of a study of freshwater lakes in Malaysia in which the dominant isolate was *A. veronii*, but they found *A. jandaei* to be the second most dominant species; however, *rpo*D sequencing was used for identification [36]. Among the *Aeromonas* spp. isolated from clinical cases, 95.4% were identified as containing only four species: *Aeromonas caviae* (37.26%), *Aeromonas dhakensis* (23.49%), *Aeromonas veronii* (21.54%), and *Aeromonas hydrophila* (13.07%) [37]. We also isolated one species of *A. dhakensis* from pond water, which was originally isolated from the stool of children with diarrhea and is considered more virulent than *A. veronii*, *A. caviae*, and *A. hydrophila* [38]. The *A. veronii* species is divided into two biovars, bv. *sobria* and bv. *veronii*, and the latter is considered more pathogenic [3]; however, in this study, we did not distinguish between *A. veronii* biovars.

The discrepancies in phenotypic and genotypic identification schemes at the species level complicate the taxonomy of the genus *Aeromonas*. Housekeeping genes such as *gyr*B and *rpo*D, which encode proteins related to DNA processing, are discriminatory phylogenetic markers, in contrast to the poor taxonomic resolution mediated by 16S rRNA in *Aeromonas* spp. [39]. Our study has the limitation of using only *gyr*B sequencing as a genetic marker for species identification; however, *gyr*B is the most frequent housekeeping gene used in the identification of motile aeromonads [39]. The *gyr*B gene has enough variable nucleotides and conserved positions and unique genetic codon usage, making it useful for species identification in *Aeromonas* spp. [40]. Furthermore, recent work suggested that even whole-genome-based multilocus phylogenetic analysis (MLPA) using six housekeeping genes may not be robust enough to identify certain strains of *Aeromonas* spp. Moreover, genomic approaches such as average nucleotide identity (ANI), in silico DNA—DNA hybridization (isDDH), and core genome phylogeny are essential for the correct identification of ambiguous *Aeromonas* strains [41].

The *A. veronii* strains detected in village aquaculture pond waters formed varying phylogroups on the basis of the *gyr*B gene. This indicates that strains have discernible variations and complex evolutionary histories. Tekedar and coworkers [42] reported a core genome phylogenetic tree for the 53 *A. veronii* genomes, which was characterized by the formation of two major clades further divided into multiple branches. The phylogenetic analysis in our study indicates the diversity of *A. veronii* isolates in village pond waters, which may belong to different ecological niches.

*Aeromonas* spp. strains are versatile opportunistic pathogens, in part complemented by multiple virulence-related factors [3]. In our study, various combinations of virulence factors within different strains of the same species were observed. On average, enolase (*eno*, 97.9%) was the most frequently detected virulence gene in the *Aeromonas* species isolated in our study, which is consistent with its almost complete prevalence reported previously [43] and high distribution (70%), as shown in another study [36]. In addition to its role in cell metabolism, enolase is involved in microbial diseases and autoimmunity [44], and promotes pathogen interactions, bacterial colonization, and pathogenesis [44,45]. In addition, the frequency of the cytotoxic enterotoxin gene encoding *A. sobria* hemolysin (act, 88.7%) was also high, and our findings corroborate those of others (ranging from 54 to 97%) [36,46,47]. Martino and coworkers [43] reported that the *act* and *asa*1 genes were present in all of their *Aeromonas* strains except *A. media*-*A. caviae*, similar to the high positivity exhibited by the *Aeromonas* isolates in our study. Cytotoxic enterotoxin (*act*) is a crucial virulence factor of this bacterium due to its association with hemolytic, cytotoxic, and enterotoxic nature [48]. Our estimates of gene *asc*V-positivity (51.6%) agree with the frequency (53.8%) determined by Rather and coworkers [47], whereas they contradict those of Khor and coworkers [36], who reported the complete absence of the *asc*V gene in their *Aeromonas* isolates. These findings demonstrate the pathogenic potential and one health importance of aeromonads prevalent in pond waters of Haryana (India), however, given the high genetic diversity of genus *Aeromonas* spp., and diversity within *Aeromonas veronii* isolates, as evidenced by phylogeny in our study, the approach of measuring frequency of the virulence gene by PCR may not be specific. A genomic approach for ascertaining virulence genes has been proposed (25). Although the role of *Aeromonas* spp. as a diahhoreal pathogen is uncertain, certain strains of *Aeromonas* may be implicated [37].

The heat-stable cytotonic enterotoxin gene *ast* was not detected in any of the 97 isolates, which is consistent with the absence of this gene in other studies [13], while some studies reported the presence of this gene, although the frequency was low (1–6%) [36,46,47].

Being waterborne, *Aeromonas* strains are potential reservoirs of antimicrobial resistance genes (ARGs), as they can easily acquire and exchange certain ARGs [49]. To effectively monitor AMR in aquatic environments, numerous authors employed *Aeromonas* as an indicator organism [50]. A portion of the antimicrobial supplied (feed or water) prophylactically or therapeutically is transported through unmetabolized feces without complete breakdown [14]. These substances remain and accumulate in the environment of fish farming in high enough quantities to exert selective pressure on aquatic bacteria and act as “genetic fuel rods” or “hotspots” for the development of AMR in aquaculture systems.

All the *Aeromonas* isolates showed complete antimicrobial resistance (100%) to ampicillin (a β-lactam antibiotic). Various studies also reported similar results [16,46,51]. This is evident because aeromonads naturally produce chromosomal β-lactamases [52], indicating an abundance of the β-lactamase gene in the gene pool of microbes of the studied source [16]. Quinolones are the first-line medications advised against aeromonad infections, and the high level of quinolone resistance of *Aeromonas* spp. to these therapies found in this study is concerning. Moreover, 40.2% of the isolates were resistant to synthetic quinolone nalidixic acid (NA), which was mainly detected in *A. hydrophila* and *A. caviae*, followed by *A. veronii* and others. This finding is in line with earlier reports of plasmid- and chromosomal-mediated quinolone resistance in *Aeromonas* spp. [53]. The main elements causing resistance to quinolones (nalidixic acid) and flouroquinolones (norfloxacin) are related to chromosomal mutations in the drug target genes and plasmid-mediated quinolone resistance genes (PMQR) [54]. *Aeromonas* spp. or *Vibrio* spp. have been reported to be common environmental species detected to carry PMQR genes.

Resistance to cefoxitin, a second-generation cephalosporin, was found in 83.3% *A. hydrophila* and 57.1% *A. caviae* isolates followed by 27.1% *A. veronii* isolates and others. Resistance in *Aeromonas* spp. isolates was also observed against third-generation cephalosporins, with the most isolates resistant to cefotaxime, followed by ceftazidime, cefpodoxime, aztreonam, and ceftriaxone. Our findings corroborate those of Silva and coworkers [55] for the high resistance observed against cefoxitin (32%) attributed to carriage of the FOX gene (a member of the *amp*C gene family) [56]. A total of 10.3% of the isolates in our study produced the ESBL enzyme, which is lower than that reported in a study in Eastern India, in which 55% of *Aeromonas* spp. isolates were ESBL-positive; however, these isolates originated from bivalves sampled from sewage-fed wetlands [57]. However, this trend is a concern since the first report of plasmid-mediated ESBL resistance was from *Aeromonas* spp. [58], until a recent report, in which ESBL-resistant *A. hydrophila* strains were isolated from deep-stage infections in humans, which also exhibited carbapenem resistance [59]. In our study, as many as 6.2% of the isolates were carbapenem-resistant. The high ACBL resistance (32%) in aeromonads detected by the phenotypic method in our study matches the 34.02% rate of cefoxitin resistance, which may be due to the presence of mobile FOX AmpC beta-lactamases common in aeromonads [56]. Interestingly, although nalidixic acid resistance was present in 40.2% of the isolates, all of the isolates were fluoroquinolone-sensitive (100%), which is consistent with earlier findings [60]. In China, an *A. punctata* strain, resistant to nalidixic acid and susceptible to flouroquinolone, was isolated from wastewater [61]. Additionally, another study by Silva and coworkers [55] reported low resistance levels and greater efficiency against *Aeromonas* species isolates for amikacin (2.1%), which is within the range obtained in this study.

The isolates resistant to at least one antimicrobial agent from three or more antimicrobial drug classes used in the study were considered multidrug-resistant (MDR) isolates [17]. Multidrug resistance (MDR) was detected among the isolates, with 29.9% of the *Aeromonas* isolates being MDR. Of those, 7.2% of the isolates showed resistance to 5 or >5 tested antimicrobials, including an *A. veronii* isolate that was resistant to 11 antimicrobials and sensitive only to tetracycline, amoxiclav, amikacin, and norfloxacin. Isolates with a MAR index of 0.2 or greater than 0.2 are considered potential high-risk sources for the spread of drug-resistant strains [51]. A total of 29.9% of the isolates in our study had a MAR index greater than 0.2, indicating that they came from a source of contamination with a high risk of contamination where antimicrobials are often employed, all of which displayed MDR. The current study’s MAR index range (0.07 to 0.73) is also more comprehensive than that of earlier studies. The MAR index in aquaculture-borne aeromonads has been reported by [51] to range from 0.12 to 0.88.

Our findings support earlier research indicating that MDR *Aeromonas* strains are spreading in the environment [13,51]. This also has important health implications since the pond water used for aquaculture is sourced through canal waters, which might carry aeromonads through various sources and transmission routes, e.g., fertilizers of fecal origin, irrigation and surface water, and water for aquaculture [62].

## 4. Materials and Methods

### 4.1. Sampling of Ponds and Isolation of Aeromonas spp.

#### 4.1.1. Sources of Samples and Sample Collection

The village ponds, which are also used as reservoirs for pisciculture by village farmers, were used as sources of water samples. One hundred pond water samples were collected from 66 villages distributed in 4 blocks under the jurisdiction of the Hisar district (Haryana, India). Each sample was collected from a single pond. All the samplings were carried out in June to September 2020 during the early morning hours. Sub-superficial water samples were collected in undisturbed waters using 500 mL sterile bottles at a distance of 2–3 m from the pond bank and at a depth of 25–30 cm below the surface water level. After sampling, the surface of the bottles was immediately sterilized using a 70% alcohol swab. The collected samples were transported immediately to the laboratory on ice and stored at refrigeration (4 °C) until use. The samples were processed for bacterial isolation on the same day or within 24 h of sample collection.

#### 4.1.2. Physicochemical Properties of Water

The pond water temperature was determined at the pond water site itself. For the determination of salinity, pH, DO, TDS, and EC, water samples were taken to the laboratory on ice within 1 h after collection and analysis. Salinity was measured with a handheld 0–32% salinity Brix refractometer with automatic temperature compensation (ATC). Total dissolved solids and pH were obtained using a TDS meter (Wellon digital TDS meter, Gujarat, India) with the electrode rinsed in distilled water before measurements to avoid any error. Conductivity was measured using a Labtronics Microprocessor COND-TDS-SAL Meter LT-51 (Haryana, India) (Siemens/centimeter, µS/cm). The dissolved oxygen content was determined using an Aquasol Dissolved oxygen test kit (North Tonawanda, NY, USA) (Code AEDO8).

#### 4.1.3. Bacteriological Examination of Samples and Isolation of Aeromonas Species

For the selective isolation and enumeration of *Aeromonas* species, starch ampicillin agar (SAA) containing ampicillin, which is based on the color change due to acid production during fermentation caused by the growth of *Aeromonas* species, was used as the selective principle [26]. The pond water samples were subjected to serial decimal dilutions up to the fifth dilution (10^−5^) in chilled sterile phosphate-buffered saline (PBS) (pH = 7.2) in duplicate. The Petri plates were incubated at 28 °C for 18–24 h. Yellow to honey-colored colonies, which are typically 3–5 mm in diameter and presumed to be *Aeromonas* spp., were used for enumeration. These colonies showed an amylase-positive reaction (surrounded by a clear zone) upon flooding with 5 mL of Lugol iodine solution [26]. Morphologically representative isolates were selected and isolated as putative members of the genus *Aeromonas* and cryopreserved in nutrient broth (NB) supplemented with 80% glycerol (*v*/*v*) at −80 °C until further analysis.

Each of the isolates was given an identification tag by prefixing “Aq” followed by a digit, which represented the sample number. If more than 1 isolate was selected from any given sample, e.g., any morphologically variant colony, then it was represented by an additional alphabet in the capitals, such as A, B, etc., from the same sample, i.e., Aq1, Aq1A, Aq2, Aq2A, etc.

### 4.2. Phenotypic and Molecular Identification

#### 4.2.1. Biochemical Testing

Each of the typical presumptive aeromonad colony types on the SAA plates was subjected to Gram staining and examined microscopically for shape and Gram reaction (Nikon Eclipse Microscope, Shinagawa, Japan). All Gram-negative rod-shaped isolates were sub-cultured on nutrient agar (NA) and subjected to cytochrome oxidase and catalase tests. Of the total representative Gram-negative, oxidase-, and catalase-positive colonies, 100 isolates were randomly selected for phenotypic confirmation, ensuring that from each pond water sample, one isolate was selected. Furthermore, the motility, utilization of citrate, malonate, glucose, lactose, and sucrose, hydrogen sulphide production, gas production from D-glucose, hemolysis on sheep blood agar (SBA), indole production, urease production, and oxidative–fermentative properties of the selected 100 putative *Aeromonas* isolates were tested via these secondary phenotypic tests. The strains were biochemically identified according to the biochemical scheme given previously [1].

#### 4.2.2. Molecular Identification by *gyr*B Gene Sequencing

The genomic DNA from biochemically identified presumptive *Aeromonas* species (*n* = 100) was isolated using a Quick-DNA Fungal/Bacterial Miniprep Kit (Zymo Research, Irvine, CA, USA) according to the manufacturer’s instructions. A PCR-amplified product (approximately 1100 bp fragment) of the *gyr*B gene was obtained using the following primers-*gyr*B_F: GGGGTCTACTGCTTCACCAA and *gyr*B_R: CTTGTCCGGGTTGTACTCGT [9]. The amplification reaction consisted of 12.5 µL of 2× DreamTaq Green PCR Master Mix (Thermo Scientific, Waltham, MA, USA), 0.2 µL of 0.4 µM of each primer, and 2 µL of genomic DNA (~250 ng/µL) and was adjusted with nuclease-free water (Thermo Scientific) in a final reaction volume of 25 µL. The reaction was carried out in a PeqSTAR 2× Universal Gradient thermal cycler, Erlangen, Germany. The selection of primers, PCR conditions, and sequencing were performed as described in a previous report [9]. Amplified products were purified using a DNA Clean and ConcentratorTM-5 Kit (Zymo Research, Irvine, CA, USA) before Sanger sequencing. The sequencing results were compared in a BLAST homology search with *Aeromonas* gene sequences submitted in the GenBank database.

#### 4.2.3. Phylogenetic Analysis of *A. veronii* Isolates

For evolutionary analysis, corresponding *gyr*B sequences of geographically distinct *A. veronii* strains (*n* = 19) were retrieved from NCBI. Evolutionary analyses were conducted using MEGA11 (https://www.megasoftware.net, accessed on 20 October 2023). The evolutionary distances were computed using the maximum composite likelihood method. The percentage of replicate trees in which the associated taxa clustered together in the bootstrap test (1000 replicates) was shown next to the branches. The tree was drawn to scale, with branch lengths in the same units as those of the evolutionary distances used to infer the phylogenetic tree. All ambiguous positions were removed for each sequence pair (pairwise deletion option). These distances were computed using the maximum composite likelihood method and are expressed in terms of the number of base substitutions per site. The ME tree was constructed using the close neighbor interchange (CNI) algorithm at a search level of 1. The neighbor-joining algorithm was used to generate the initial tree. Codon positions included were 1st + 2nd + 3rd + noncoding, and all ambiguous positions were removed (pairwise deletion). The analysis involved 78 nucleotide sequences, with a total of 1190 positions in the final dataset. Evolutionary analyses were conducted using MEGA11 (v. 11.0.6) software. Likewise, evolutionary relationships of isolated species other than *A. veronii* were also drawn.

### 4.3. Genetic Traits of Potential Virulence Factors

The presence of 7 putative determinant genes of virulence, namely *ahh*1, *asa*1, *act*, *ast*, *asc*V, *eno*, and *aex*T, was screened in the *Aeromonas* spp. isolates through direct PCR amplification as reported previously by Martino and coworkers for determining *Aeromonas* spp. diversity from aquatic origins [43]. Details of their primers, expected product sizes, and PCR conditions are summarized in Appendix A.

The binary data obtained from the virulence gene detection patterns were used for calculating the pairwise genetic distance between isolates with the help of Nei’s standard genetic distance method [63]. Afterwards, agglomerative hierarchical clustering of the isolates was performed using Ward’s approach [64]. The number of clusters is chosen based on the Bayesian information criterion (BIC), i.e., 5. The clustering of isolates into different clusters is presented as a dendrogram (Appendix A). Clustering analysis was performed using R and RStudio software (R version 4.3.10) with the help of the packages adegenet (exploratory analysis of genetic and genomic data), NAM, circle, and dendextend [65,66,67].

### 4.4. Antimicrobial Susceptibility

Antimicrobial susceptibility assays were performed by the disc diffusion method according to a standard procedure based on the guidelines of the Clinical and Laboratory Standards Institute (CLSI) document M45-Ed3 [68] using E. coli ATCC25922 as a reference strain. A panel of 15 antimicrobials belonging to the following 8 classes were used for the assay: class {Antimicrobial name, (abbreviation and disc content in µg)}: penicillin and beta-lactams {ampicillin (AMP/10); amoxicillin-clavulanic acid (AMC, 20/10)}; phenicols {chloramphenicol (CHL/30)}; folate pathway inhibitors {trimethoprim-sulfamethoxazole (SXT, 1.25/23.75)}; cephems {ceftriaxone (CRO/30); cefpodoxime (CPD/10); ceftazidime (CAZ/30); aztreonam (ATM/30); cefotaxime (CTX/30); cefoxitin (FOX/30)}; carbapenem {imipenem (IPM/10)}; aminoglycosides {amikacin (AMK/30)}; flouroquinolone {norfloxacin (NOR/5); and quinolone {nalidixic acid (NAL/30)} and tetracyclines {tetracycline (TET/30)}. Colonies of isolates were incubated in nutrient broth (HiMedia). After achieving 0.5 MacFarland turbidity (Biosan Densitometer, Riga, Latvia), the bacterial suspension was inoculated on 100 mm diameter pre-dried Mueller Hinton Agar (MHA, HiMedia) plates using a sterile cotton swab, and immediately, the discs were transferred onto the plates. The inoculated plates were incubated at 37 °C for 24 h. According to CLSI zone diameter interpretation criteria (CLSI M45-Ed3), zones of inhibition were recorded, and isolates were classified as sensitive (S), immediate (I), or resistant (R). In addition, the multiple antibiotic resistance (MAR) index was computed by dividing the number of antimicrobials to which the isolate was resistant by the total number of antimicrobials examined [13].

Furthermore, all the aeromonads that were resistant to CRO, CPD, ATM, CAZ, CTX, and FOX were screened for extended-spectrum β-lactamase (ESBL) (*n* = 10) and AmpC β-lactamase production (ACBL) (*n* = 33) using double disc synergy methods [69]. The double discs contained ceftazidime–clavulanic acid-combined discs (CAC-30/10 µg), cefotaxime–clavulanic acid-combined discs (CEC-30/10 µg), and cefoxitin–cloxacillin-combined discs (CXX-30/200 μg) for the ESBL and ACBL assays, respectively. Following incubation for 18–24 h at 37 °C, an increase of >5 mm and >4 mm in the zone around the combined disc compared to the corresponding disc alone was considered positive for ESBL and ACBL production, respectively.

## 5. Conclusions

This study revealed a potential risk of multidrug-resistant pathogenic aeromonad populations in the surface water of aquaculture ponds in the Hisar region of Haryana (India). Aeromonads were isolated from all the pond water samples. The dominant species with enterotoxigenic potential found in this study included *A. veronii*, *A. hydrophila*, and *A. caviae*. The number of aeromonads in pond water varied between 10^2^ and 10^8^ cfu/mL. Molecular identification of *Aeromonas* spp. by *gyr*B sequence revealed that 97% of the isolates were aeromonads, with *A. veronii* being the most dominant species, followed by *A. caviae*, *A. hydrophila*, *A. jandaei*, *A. sobria*, and *A. dhakensis*. The constructed phylogeny revealed the formation of major and minor clades, indicating genotypic diversity among the *A. veronii* isolates. Carriage of multiple-virulence genes was common among the *Aeromonas* isolates, and cluster analysis by dendrogram revealed multiple distinct clusters depicting virulence-mediated heterogeneity in the species. High antimicrobial resistance was detected against ampicillin (100%), nalidixic acid (40.2%), and cefoxitin (33%). Its resistance to third-generation cephalosporins indicates that ESBL resistance (10.3%) and ACBL resistance (34.02%) are emerging problems in aeromonads. The overall MAR of 0.07 to 0.73 and the prevalence of *Aeromonas* spp. with ESBL, ACBL, and carbapenem resistance are alarming. This study highlights that aquaculture pond waters in this region are infested with potentially pathogenic MDR strains of motile aeromonads, which has implications for one health, as aquaculture is an increasingly important source of aquatic dietary protein in form of fish and shrimp culture supplies. Furthermore, the sustainable growth of this sector is dependent on the health of the aquatic pond environment system, which is also used for animal husbandry, farm irrigation, and human use. This study underlines the importance of the surveillance of freshwater bodies to assess the level of pathogenic and drug resistance genotypes of *Aeromonas* spp. and importance of infection prevention and control (IPC) measures to control MDR motile aeromonads in water bodies used for aquaculture and other economic activity.

## Figures and Tables

**Figure 1 antibiotics-14-00294-f001:**
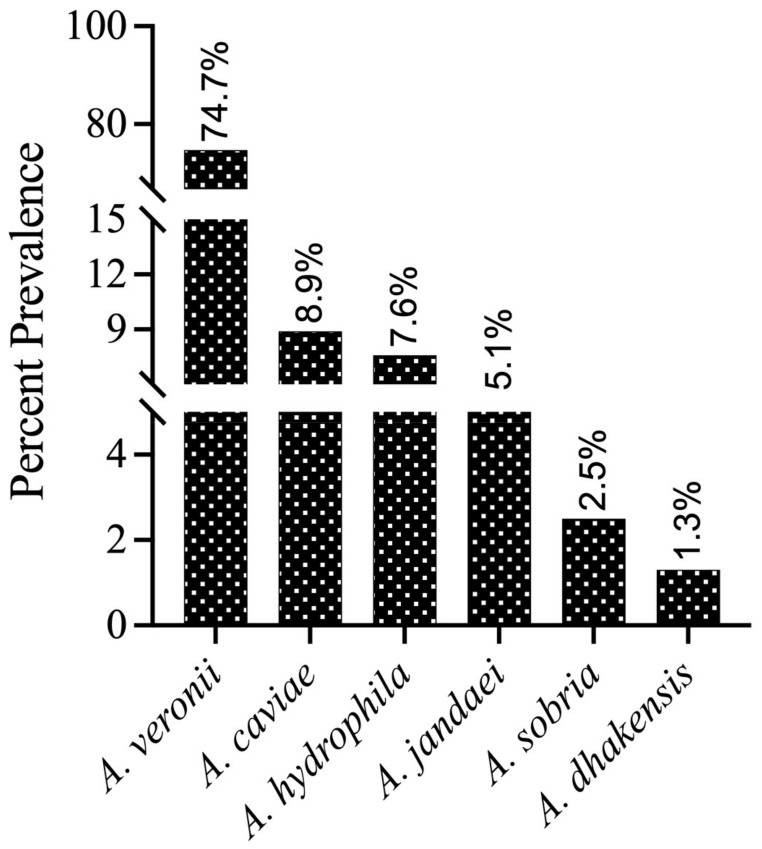
Prevalence (%) of different *Aeromonas* species in the fish culture ponds confirmed by *gyr*B gene sequencing and submitted to NCBI (*n* = 79).

**Figure 2 antibiotics-14-00294-f002:**
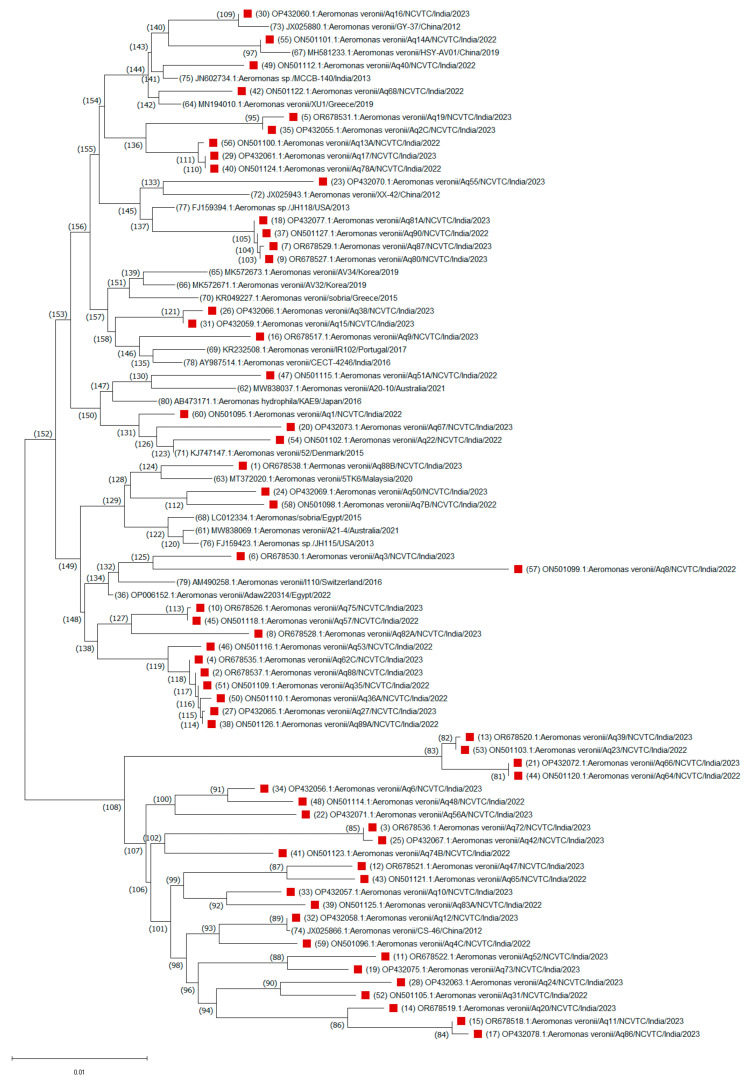
Unrooted neighbor-joining phylogenetic trees based on the *gyr*B gene sequences of a total of 79 (59 strains from this study, red blocks and the remaining are 19 strains from GenBank) *Aeromonas veronii* strains. Evolutionary history was inferred using the neighbor-joining method. The percentages of replicate trees in which the associated taxa clustered together based on 1000 bootstrap replicates are shown adjacent to branches. Sequence accession numbers are in parentheses.

**Figure 3 antibiotics-14-00294-f003:**
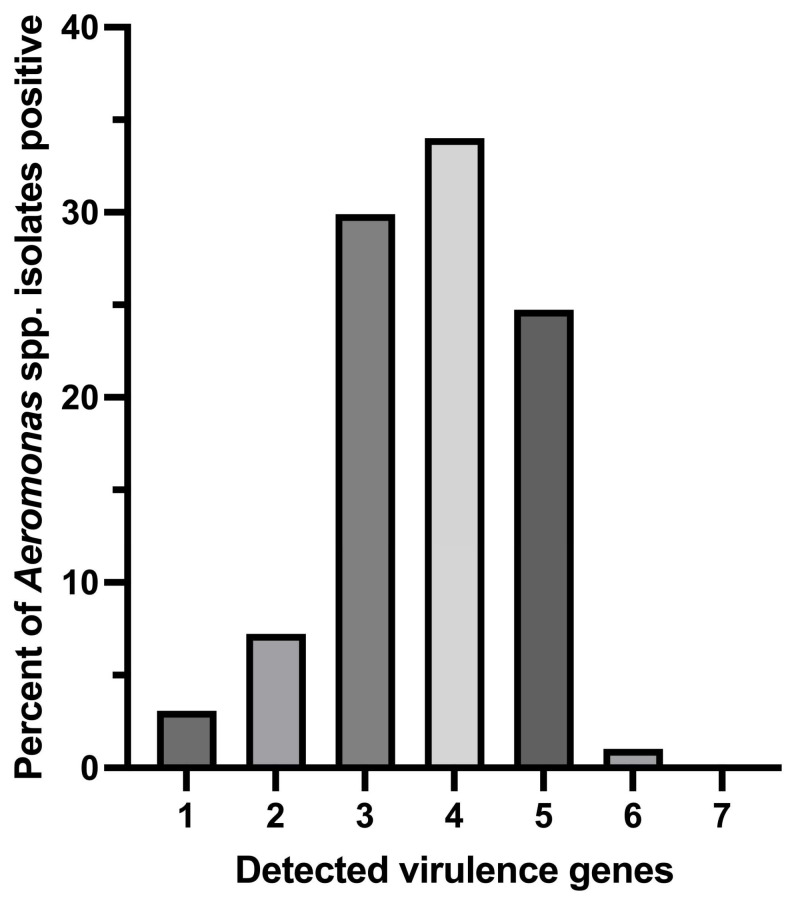
Distribution of multiple virulence gene indices of *Aeromonas* species isolates positive for tested virulence genes.

**Figure 4 antibiotics-14-00294-f004:**
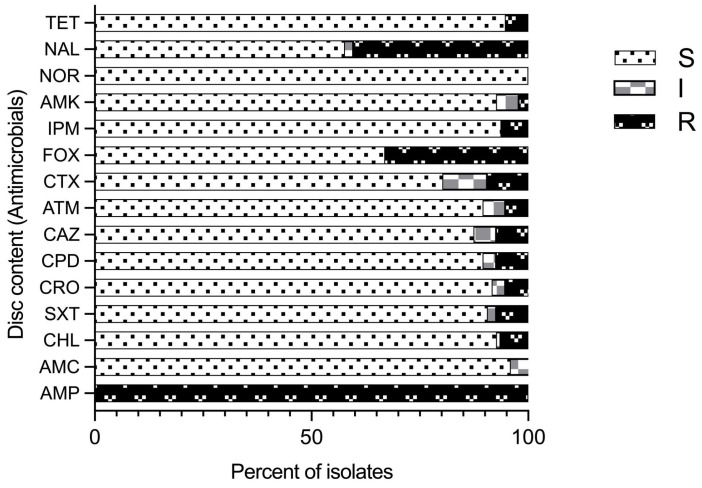
Antimicrobial susceptibility pattern for *Aeromonas* species isolates for different antimicrobials tested and their classes: tetracycline (TET); nalidixic acid (NAL); norfloxacin (NOR); amikacin (AMK); imipenem (IPM); cefoxitin (FOX); cefotaxime (CTX); aztreonam (ATM); ceftazidime (CAZ); cefpodoxime (CPD); ceftriaxone (CRO); trimethoprim-sulfamethoxazole (SXT); chloramphenicol (CHL); amoxicillin-CA (AMC); and ampicillin (AMP); (R–resistant; I–intermediate; and S–sensitive).

**Figure 5 antibiotics-14-00294-f005:**
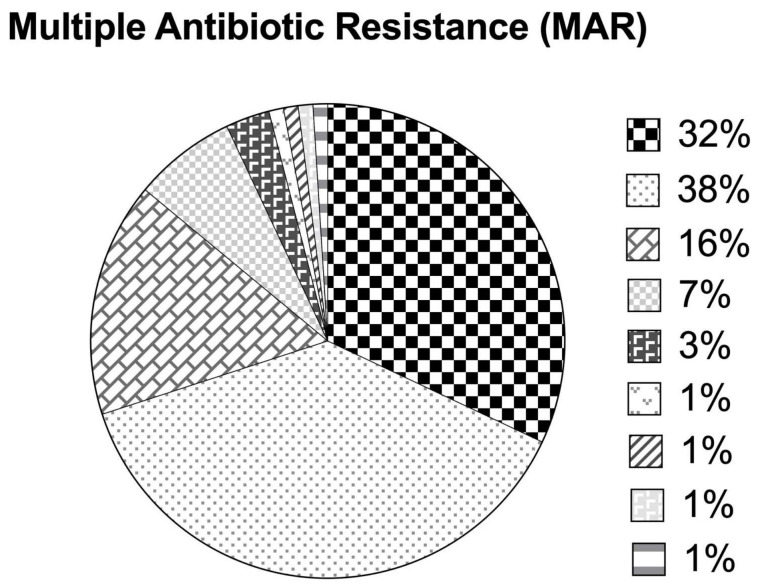
Multiple antibiotic resistance indices based on antimicrobial susceptibility patterns among *Aeromonas* species isolates.

**Table 1 antibiotics-14-00294-t001:** Physicochemical parameters of pond water samples.

ID	Temp (°C)	pH	TDS (ppm)	Salinity (ppt)	DO (mg/L)	EC (µs/cm)
Mean	32.3	7.4	643.38	0.1	6.4	960.27
Minimum	29	6.8	111	0.1	2.9	165.67
Maximum	34	9.0	2440	0.3	8	3641.79
Reference value range [23,24,25]						
HDL; acceptable range	22–35	5.5–10	500	-	>5	30–5000
MPL; desirable range	25–32	6.5–9.5	1000	-	>5	100–2000

Note: TDS—Total dissolved solids; DO—dissolved oxygen; EC—electrical conductivity; HDL—highest desirable limit; and MPL—maximum permissible limit.

**Table 2 antibiotics-14-00294-t002:** Detection frequency of different virulence genes among *Aeromonas* species isolates.

Virulence Gene	Detection Frequency (%)
*ahh*1	4.12
*asa*1	85.6
*act*	88.7
*ast*	0
*asc*V	51.6
*eno*	97.9
*aex*T	45.4

## Data Availability

Nucleotide sequence data of the *Aeromonas* spp. isolates are accessioned and available in NCBI. Details are available in Appendix A.

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
