# Peer review of "Aeromonas Species Diversity, Virulence Characteristics, and Antimicrobial Susceptibility Patterns in Village Freshwater Aquaculture Ponds in North India"

_antibiotics, 2025, doi:10.3390/antibiotics14030294_

Round 1
Reviewer 1 Report
Comments and Suggestions for Authors
The reviewed work explores the diversity, virulence factors, and antimicrobial resistance patterns of Aeromonas species in village aquaculture ponds in North India, addressing a significant public and animal health concern. While the study's methodology is robust and the findings are relevant, critical issues related to data presentation, classification errors, and result interpretation need substantial improvement. Specifically, percentages are inconsistently reported without absolute values, cefoxitin is misclassified, and dense sections would benefit from clearer data visualization. Additionally, deeper contextualization of antimicrobial resistance findings is necessary. Due to these substantial concerns, I recommend major revisions. Addressing these issues will significantly improve the manuscript's clarity, scientific accuracy, and overall impact.
Comments on the Quality of English Language
A thorough English language editing by a native speaker or a professional language editing service is strongly recommended to enhance the manuscript's clarity, flow, and scientific presentation.
Author Response
The reviewed work explores the diversity, virulence factors, and antimicrobial resistance patterns of Aeromonas species in village aquaculture ponds in North India, addressing a significant public and animal health concern. While the study's methodology is robust and the findings are relevant, critical issues related to data presentation, classification errors, and result interpretation need substantial improvement.
Specifically, percentages are inconsistently reported without absolute values,
Resp reviewer, thank you for your comment. Sir, we would like to mention that in abstract, the absolute numbers has not been mentioned due to need of brevity, and journal requirement of limitation to 150 words, however same has been shown in Results (Line 154-156). Similarly, the absolute numbers are given in section for antimicrobial resistance also in Line 261-263. For AMR, absolute values are available in the supplementary table 4. For biochemical test positivity percentages, values are available in Supplementary Table 2. For ESBL & ACBL, numerical values are given (L277-283. Pg10).
However, as per your suggestion, we have now added absolute values for section 2.3 Virulence gene characteristics L 203-208.
cefoxitin is misclassified, and
Respected Reviewer, with regard to classification of cefoxitin, we have noted your comment and added its being a cephamycin. (L 265).
dense sections would benefit from clearer data visualization.
Respected Reviewer, we agree with your observation. We have gone through the MS and have brought changes for clarity.
Additionally, deeper contextualization of antimicrobial resistance findings is necessary.
Sir, we fully agree with your apt observation. Since this is very important, we have added some discussion in antimicrobial resistance findings.
We have, as result of your observation, added 2 references (Ref No 71, 72).
Changes have also been brought about by insight of other reviewers.
Due to these substantial concerns, I recommend major revisions. Addressing these issues will significantly improve the manuscript's clarity, scientific accuracy, and overall impact.
Sir, thank you for your comments and pointing out of flaws. Your constructive comments have added to the manuscript's clarity, scientific accuracy, and overall impact, for which we are greatly thankful.
Comments on the Quality of English Language
A thorough English language editing by a native speaker or a professional language editing service is strongly recommended to enhance the manuscript's clarity, flow, and scientific presentation.
Resp, we have gone through the language of the manuscript thoroughly, please.
Submission Date
19 December 2024
Date of this review
11 Jan 2025 14:03:46
Reviewer 2 Report
Comments and Suggestions for Authors
Abstract
l. 24-25: the names of genes need to be explained, i.e. what they encode.
l. 29-30: A. veronii is repeated twice in this sentence
The Introduction is very well written. All information necessary for understanding why the study was undertaken is provided and the relevant references are cited.
The aim is well justified. I only suggest adding a sentence or two that would explain the actual outcomes of the study in terms of one health perspective. Will the study contribute to expanding our general knowledge on the human and environmental health? I would say yes, but this needs to be emphasized in the last paragraph of the Introduction.
Results
The scope of research was really broad. I do not have any major substantial remarks, just some technical minor issues, like e.g. Figure 4 is placed in the middle of the sentence which makes reading a bit confusing as it seems like the sentence was interrupted.
Discussion
my remarks are also minor and technical:
l. 312, 313, 314, 316: the CFU values are presented as e.g. (101 -103 CFU/ml). I assume that these should be written with upper index.
Another technical remark – the italics is missing in some places (e.g. Aeromonas or gyrB).
The Materials and Methods section lacks specified information about statistical analysis. Please add a subchapter that describes the methods and software used to test the statistics.
Conclusions
l. 591: the unit is missing next to the number of aeromonads.
l. 603-605: This sentence should introduce some more specific information about the implications the obtained results have for the One Health. Please add a few sentences that would elaborate more on the specific outcomes of this research. This is important because the fragment in lines 587-602 is just a summary of the results, while this section should provide the actual conclusions.
Apart from these remarks, I do not have any other. This research is super interesting, relevant to the current health-related and environmental trend of emerging pollutants/contaminants. The study is well designed, the methods used are up to date and described with all details or references that would allow to reproduce this study.
Author Response
-
Comments and Suggestions for Authors
Abstract
l24-25: the names of genes need to be explained, i.e. what they encode.
Resp reviewer, we have not mentioned these in abstract because these were adding to word limit. However, now we have added these.
- 29-30: A. veronii is repeated twice in this sentence
Thank you for your comment! Actually, these are two subsets of A. veronii strains. 59 from our study and rest 19 from GenBank. A word together has been added in order to clarify this.
The Introduction is very well written. All information necessary for understanding why the study was undertaken is provided and the relevant references are cited.
The aim is well justified. I only suggest adding a sentence or two that would explain the actual outcomes of the study in terms of one health perspective. Will the study contribute to expanding our general knowledge on the human and environmental health? I would say yes, but this needs to be emphasized in the last paragraph of the Introduction.
Resp Reviewer, this is in fact a very insightful suggestion. We have inserted a small paragraph (L93-101) on this aspect. ‘’ Our study highlights the importance of inland aquaculture pond systems as the ecological compartments of environment, which on account of their role as endpoint of organic wastes from human and animals, may act as local epicenters of AMR evolution and dissemination, affecting the integrity of food-chain, and animal and human health’’.
Results
The scope of research was really broad. I do not have any major substantial remarks, just some technical minor issues, like e.g. Figure 4 is placed in the middle of the sentence which makes reading a bit confusing as it seems like the sentence was interrupted.
Your point is very valid. The figure has been placed now appropriately.
Discussion
my remarks are also minor and technical:
- 312, 313, 314, 316: the CFU values are presented as e.g. (101 -103 CFU/ml). I assume that these should be written with upper index.
Thank you for pointing out the mistake. These have been taken care now.
Another technical remark – the italics is missing in some places (e.g. Aeromonas or gyrB).
Resp. Reviewer, we are sorry for the omissions. Corrections has been made on keywords & line 488, 535, 648; Line 520 for gyrB.
The Materials and Methods section lacks specified information about statistical analysis. Please add a subchapter that describes the methods and software used to test the statistics.
Resp. sir, no specific statistical analysis was needed. Specific software for evolutionary analysis (MEGA) has been mentioned in section. The Nei's standard genetic distance method, and Ward's approach references have been added. Clustering of isolates based on Bayesian information criterion (BIC), for dendrogram is mentioned. Clustering analysis software R and RStudio software with the packages adegenet (Exploratory Analysis of Genetic and Genomic Data), NAM, circle and dendextend has been mentioned with reference.
Conclusions
- 591: the unit is missing next to the number of aeromonads.
Resp reviewer, thank you for pointing out the omission. Unit of cfu/ml has been inserted.
- 603-605: This sentence should introduce some more specific information about the implications the obtained results have for the One Health. Please add a few sentences that would elaborate more on the specific outcomes of this research. This is important because the fragment in lines 587-602 is just a summary of the results, while this section should provide the actual conclusions.
Resp reviewer, thank you for your insight and suggestion. Following paragraph has been added
………as aquaculture is an increasingly important source of aquatic dietary protein in form of fish and shrimp culture supplies. Furthermore, the sustainable growth of this sector is dependent on health of aquatic pond environment system, which in turn are also used for animal husbandry, farm irrigation and human use’’. See Line 637-640.
Apart from these remarks, I do not have any other. This research is super interesting, relevant to the current health-related and environmental trend of emerging pollutants/contaminants. The study is well designed, the methods used are up to date and described with all details or references that would allow to reproduce this study.
Respected, reviewer, thank you so much for your kind words. Your comments have motivated us!
Submission Date
19 December 2024
Date of this review
17 Jan 2025 10:28:48
Reviewer 3 Report
Comments and Suggestions for Authors
The manuscript authored by Nokhwal et al. investigates the diversity, virulence characteristics, and antimicrobial resistance (AMR) patterns of Aeromonas species isolated from freshwater aquaculture ponds in North India. The study identifies a high prevalence of multidrug-resistant A. veronii, A. hydrophila, and A. caviae, emphasizing the significant public health and environmental risks these pathogens pose and the urgent need for improved aquaculture management and AMR surveillance. The topic is highly relevant to the scope of Antibiotics, as freshwater aquaculture is a recognized reservoir and source of antibiotic resistance. By sampling multiple aquaculture ponds, the study presents findings that are broadly representative of the studied region. The manuscript is well-written and relatively easy to follow, but it requires substantial revisions before being suitable for acceptance.
Major Comments:
-
Novelty and Impact: While the study is significant, the introduction does not clearly articulate its novelty or impact. Please specify what research gaps this study addresses and how it advances prior knowledge. For instance, what specific questions were targeted in this study, and what novel insights were gained compared to previous research?
-
Water Quality Parameters: Although the water quality parameters provided in Section 2.1 are informative, their relevance to the study’s main results (e.g., Aeromonas strains, AMR profiles, or virulence factors) is unclear. Please justify the inclusion of this section in the main results or clarify how these parameters contribute to understanding the findings.
-
Scientific Notation Formatting: Superscript formatting for scientific notation is inconsistent. For example, lines 121, 312, 313, 314, and 316. Correct these instances and ensure uniform formatting throughout the manuscript.
-
Phylogenetic Analysis: For Figure 2, are the 19 Aeromonas strains from GenBank the only available strains? If not, provide a rationale for selecting these specific strains for comparison. Address how these choices influence the interpretation of phylogenetic relationships.
-
Selection of Virulence Genes: Why did the study target only seven virulence genes? Are these the most relevant or commonly studied genes in Aeromonas species? Justify their selection and explain whether other potentially important genes were excluded.
-
Figure 4 Axis Scale: The "percent of isolates" scale in Figure 4 exceeds 100, which is not logically possible. Revise the scale to a maximum of 100 to ensure clarity and accuracy.
Author Response
Comments and Suggestions for Authors
The manuscript authored by Nokhwal et al. investigates the diversity, virulence characteristics, and antimicrobial resistance (AMR) patterns of Aeromonas species isolated from freshwater aquaculture ponds in North India. The study identifies a high prevalence of multidrug-resistant A. veronii, A. hydrophila, and A. caviae, emphasizing the significant public health and environmental risks these pathogens pose and the urgent need for improved aquaculture management and AMR surveillance. The topic is highly relevant to the scope of Antibiotics, as freshwater aquaculture is a recognized reservoir and source of antibiotic resistance. By sampling multiple aquaculture ponds, the study presents findings that are broadly representative of the studied region. The manuscript is well-written and relatively easy to follow, but it requires substantial revisions before being suitable for acceptance.
Resp reviewer, thank you for your words and critical evaluation.
Major Comments:
Novelty and Impact: While the study is significant, the introduction does not clearly articulate its novelty or impact.
Please specify what research gaps this study addresses and how it advances prior knowledge. For instance, what specific questions were targeted in this study, and what novel insights were gained compared to previous research?
Resp. Reviewer, we are thankful for your highlighting the novelty and impact aspects to be included articulately in Introduction. The most novel and impactful aspect of the study has been mentioned in discussion (L308-310) that, this study, for the first time reports the occurrence and identification of aeromonads in village aquaculture ponds in the sampled region of Haryana (India). It thus first time adds to the evidence of status of AMR in aquaculture realm of environment in this region of world.
On your suggestion, the same has now been rephrased in Introduction, with additional comment about addressing one of the objectives of AMR Global action plan, i.e., the need of `gathering of evidence base through surveillance and research’.
The world is in the midst of a silent pandemic of antimicrobial resistance. The Global Action Plan on AMR has emphasized gathering of evidence of antimicrobial resistance base through surveillance and research. This study, aims to address the aquatic environmental surveillance on AMR in village pond waters used for aquaculture, animal husbandry and human in the northern region of Haryana (India) for the first time.
Water Quality Parameters: Although the water quality parameters provided in Section 2.1 are informative, their relevance to the study’s main results (e.g., Aeromonas strains, AMR profiles, or virulence factors) is unclear. Please justify the inclusion of this section in the main results or clarify how these parameters contribute to understanding the findings.
Resp. Reviewer, thanks for your comments. Our study on water quality parameters is relevant to this study as these village ponds are being used for freshwater aquaculture, as highlighted in the title. Your observation on relevance of this section is apt in reference to isolation of Aeromonas spp, however, the water quality parameters were within range for freshwater aquaculture (See in Discussion, 324-332). Since, as per workers, water temperature and salinity has influence on total aeromonads in aquatic environment and also on virulence of A. hydrophila.
Paper on influence of temperature: Sadique A, Neogi SB, Bashar T, Sultana M, Johura FT, Islam S, Hasan NA, Huq A, Colwell RR, Alam M. Dynamics, Diversity, and Virulence of Aeromonas spp. in Homestead Pond Water in Coastal Bangladesh. Front Public Health. 2021 Jul 9;9:692166. doi: 10.3389/fpubh.2021.692166. PMID: 34307285; PMCID: PMC8298834.
Paper on influence of water salinity: (John N, Vidyalakshmi VB, Hatha AAM. Effect of pH and Salinity on the Production of Extracellular Virulence Factors by Aeromonas from Food Sources. J Food Sci. 2019 Aug;84(8):2250-2255. doi: 10.1111/1750-3841.14729. Epub 2019 Jul 17. PMID: 31313323.)
Your comment
Scientific Notation Formatting: Superscript formatting for scientific notation is inconsistent. For example, lines 121, 312, 313, 314, and 316. Correct these instances and ensure uniform formatting throughout the manuscript.
Resp. reviewer, thank you for pointing out the mistakes. The scientific notation has been corrected.
Phylogenetic Analysis: For Figure 2, are the 19 Aeromonas strains from GenBank the only available strains? If not, provide a rationale for selecting these specific strains for comparison. Address how these choices influence the interpretation of phylogenetic relationships.
Sir, indeed the 19 Aeromonas spp. strains were not the only one available in GenBank, however, we chose these strains because we believe that 19 strains (32%) originating from all the continents of the world (USA, Europe, Australia, Asia) to be compared with 59 strains from this study will result in a robust, yet manageable tree to inform us of biodiversity of Aeromonas veronii strains, which were the most abundant species in this study.
Selection of Virulence Genes: Why did the study target only seven virulence genes? Are these the most relevant or commonly studied genes in Aeromonas species? Justify their selection and explain whether other potentially important genes were excluded.
Resp. Reviewer, you have rightly pointed out that these 7 virulence genes were targeted as these are the common genes which are surveyed for delineating the virulence capabilities of motile aeromonads, specifically in as is evidenced from literature. We have followed Martino et al, (Martino ME, et al Determination of microbial diversity of Aeromonas strains on the basis of multilocus sequence typing, phenotype, and presence of putative virulence genes. Appl Environ Microbiol. 2011 Jul;77(14):4986-5000. doi: 10.1128/AEM.00708-11. Epub 2011 Jun 3. PMID: 21642403; PMCID: PMC3147379.) This is also mentioned in Section 4.3 material and methods.
However, given the importance of your point, we have added the following sentence in discussion, which has added depth to this discussion.
…given the high genetic diversity of genus Aeromonas spp, and diversity within Aeromonas veronii isolates, as evidenced by phylogeny in our study, the approach of measuring frequency of virulence gene by PCR may not be specific. A genomic approach for ascertaining virulence genes has been proposed (25). (See line 404-408).
Figure 4 Axis Scale: The "percent of isolates" scale in Figure 4 exceeds 100, which is not logically possible. Revise the scale to a maximum of 100 to ensure clarity and accuracy.
Resp. Reviewer, thank you for pointing out this mistake. We have corrected the figure, and a revised figure has been submitted.
Round 2
Reviewer 1 Report
Comments and Suggestions for Authors
I have no further commments
Comments on the Quality of English Languagenothing to add